# Measles vaccines and non-specific effects on mortality or morbidity: A systematic review and meta-analysis

Louise A. Fournais[1,2☯*], Anne C. Zimakoff[1☯], Andreas Jensen[1‡],
Jeppe H. Svanholm[3,4‡], Ingvild Fosse[3,5‡], Lone G. Stensballe[1,3‡]

1 The Child and Adolescent Department, The Danish National University Hospital, Copenhagen, Denmark, 2 The Child and Adolescent Department, Hvidovre University Hospital, Hvidovre, Denmark, 3 Copenhagen University, Faculty of Medical and Health Sciences, Copenhagen, Denmark, 4 Department of Biochemistry, Hvidovre University Hospital, Hvidovre, Denmark, 5 Department of Gynaecology and obstetrics, Aalborg University Hospital, Thisted Department, Thisted, Denmark

☯ These authors contributed equally to this work.
‡ These authors also contributed equally to this work.
* louisefournais@hotmail.com

## Abstract

### Background

This review evaluates the hypothesis of beneficial non-specific effects of the standard-titre measles vaccine.

### Methods

We conducted a systematic review and meta-analysis of randomised controlled trials. Trials included standard or high-titre live attenuated measles containing vaccines compared to other vaccines or placebo. The primary outcomes were mortality and morbidity. Secondary outcomes were infections, antibiotic use, atopy, allergies, asthma, and atopic dermatitis.

### Findings

23 articles were included in this systematic review.

Mortality: Two doses of measles vaccine vs. only one dose showed no significant effect on mortality; risk ratio (RR) = 1.05 (95% CI: 0.78 to 1.41), p=0.76. The analysis was based on a relative risk reduction (RRR) of 25% and a control group event rate of 2.32% as measured in the actual trials included in the analysis. In males, the association was rejected: RR = 1.09 (0.86 to 1.37), p=0.47. In females, the association was not rejected at 25%, but was at 33% level: RR = 1.0 (0.64 to 1.54), p=0.99.
Morbidity: Overall, the hypothesis was rejected: RR = 0.94 (0.80 to 1.10), p=0.43. The rejection was sustained for both sexes: females RR = 0.95 (0.77 to 1.18), p=0.6; males RR = 0.92 (0.83 to 1.03), p=0.13.

**Editor:** Ahmad Khalid Aalemi, University of Manchester School of Biological Science: The University of Manchester Faculty of Biology Medicine and Health, UNITED KINGDOM OF GREAT BRITAIN AND NORTHERN IRELAND

**Data availability statement:** All relevant data are within the paper and its Supporting Information files.

**Funding:** The author(s) received no specific funding for this work.

**Competing interests:** The authors have declared that no competing interests exist.

## Interpretation

Based on evidence from randomised controlled trials, this systematic literature review and meta-analysis did not support the hypothesis of non-specific effects of standard-titre measles containing vaccines. Trial Sequential Analysis indicated that the meta-analysis included sufficient data to reach this conclusion.

## Trial registration

PROSPERO CRD42022344473

## Introduction

Measles is an acute viral disease causing about 134,200 deaths yearly [1]. Measles vaccines are effective; but due to the contagiousness of the virus, ≥95% must be vaccinated to control transmission [2]. First doses are usually given at 12 months of age, or from 9 months in endemic settings [3].

In the late 1970s, observational studies in low-income countries revealed an unexpected decline in overall child mortality among measles-vaccinated children that surpassed the anticipated reduction in measles-related deaths [4]. This led to the hypothesis that live-attenuated measles containing vaccines (MCVs) have beneficial non-specific effects. Subsequently randomised controlled trials (RCTs) tested the safety and efficacy of standard-titre (STMV) and high-titre measles-containing vaccines (HTMV) in Senegal, Haiti, Sudan, and Guinea Bissau in the late 1980s and 1990s [5–8]. These trials showed increased long-term mortality, especially in females, after HTMV [9]. Consequently, HTMVs were withdrawn from child immunisation programmes in 1992 [10].

Non-specific effects of vaccines are defined as health effects reaching beyond protection against the target pathogen. The hypothesis of beneficial non-specific effects of MCVs has been investigated for more than 30 years. Prior studies suggested that live-attenuated vaccines may reduce child mortality, morbidity and improve growth [11–13], while inactivated vaccines may worsen these outcomes [14,15]. While numerous non-randomised studies proposed such effects [13,14], statistically significant results have rarely been replicated in RCTs [11,12,16]. Thus, the existing results were ambiguous. Since MCVs are used globally in an era where vaccine hesitancy is recognized as a major global health threat [17], it is crucial to understand health effects of vaccines and clinically important to know whether the non-specific effects observed on observational studies were attributable to confounding.

This systematic review summarises, meta-analyses, and evaluate evidence on potential non-specific effects of MCVs. Focus is on morbidity and mortality after STMV, though HTMV is also considered.

## Methods

This systematic review followed the Preferred Reporting Items for Systematic Reviews and Meta-analyses (PRISMA) statement and The Cochrane Handbook

[18,19]. The protocol was registered in the International prospective register of systematic reviews (PROSPERO, CRD42022344473). As RCTs represent the highest level of evidence [18], only RCTs were included. Ethical approval was not needed since all data included in this review has already been published in other articles that had gained consent.

The co-primary outcomes were mortality and morbidity. Secondary analyses scrutinised sex-differential effects and focused on the follow-up period where infants who were randomised to early MCV were compared with control-infants not yet vaccinated with MCV (one vs. zero dose MCV).

### Systematic search and screening

A systematic search was initiated in February 2022. The search was computerised and performed on EMBASE and PubMed. The search was repeated weekly until October 2022 and confirmed in August 2023. The search string can be found in supplementary materials (S1 Table). EndNote 20 and Covidence were used to manage the identified articles. Covidence was used for the screening process. Two authors screened abstracts independently according to the PICO criteria (S1 Appendix). Discrepancies were resolved by a third author. The three authors independently screened all full texts. Disagreements were resolved by group discussion or with the senior researcher.

### Quality assessment and data extraction

Three authors read all included trials and performed independent quality assessment and data extraction. Disagreements were discussed in the group. The quality assessment was based on the ROB-2 tool, as recommended by the Cochrane Handbook [20]. (S2 Table and S1 Appendix)

### Statistics

Two different meta-analyses were performed: forest plots using R version 4.2.1 (R Core Team, Vienna, Austria) with fixed and random effects, and Trial Sequential Analyses (TSA) using the TSA software, 0.9.5.10 beta version (available from ctu.dk/tsa) [21]. TSA helps determine when enough data have been collected to draw conclusions. To avoid including the same child more than once, only data from original trials were used in the meta-analyses. The methods are further detailed in S1 Appendix.

## Results

The literature search identified 4315 unique articles. After initial screening, 4224 articles were excluded due to non-eligible interventions or study design. The full text of 92 articles were assessed, and 23 were included. Reasons for exclusions are presented in the PRISMA diagram (Fig 1) and in S5 Table. Trial characteristics are presented in supplementary materials (S1 Fig and S2 Table-S4 Table).

### The standard-titre measles vaccine

**Mortality.** Seven trials (35,516 participants) investigated mortality as the primary outcome. Five were original trials, two were re-analyses of already included trials. Four trials compared a two-dose schedule with a one-dose schedule. Three trials compared a dose at 9 months with an additional early dose at 4 months. Follow-up was until 36 months (two trials) [11,12] or 60 months (one trial) [16]. One trial compared a dose at 9 months with a booster dose at 18 months, with 48 months follow-up [22]. The fifth original trial was cluster randomised and compared a 'restrictive measles vaccine policy' with a 'measles vaccine for all policy' with 60 months follow-up [23]. Two re-analyses used data from the trials with an extra early dose and 36 months of follow-up [24,25].

Forest plots showed no differences in mortality: crude fixed effect model RR = 0·99 (95% confidence interval 0·85 to 1·16). Random effects model: RR = 1·00 (0·82 to 1·22), $I^2$ = 37% (S2 Fig). Adjusted fixed effect model: RR = 1·00 (0·84 to 1·18). Adjusted random effects model: RR = 1·00 (0·83 to 1·20), $I^2$ = 7% (Fig 2).

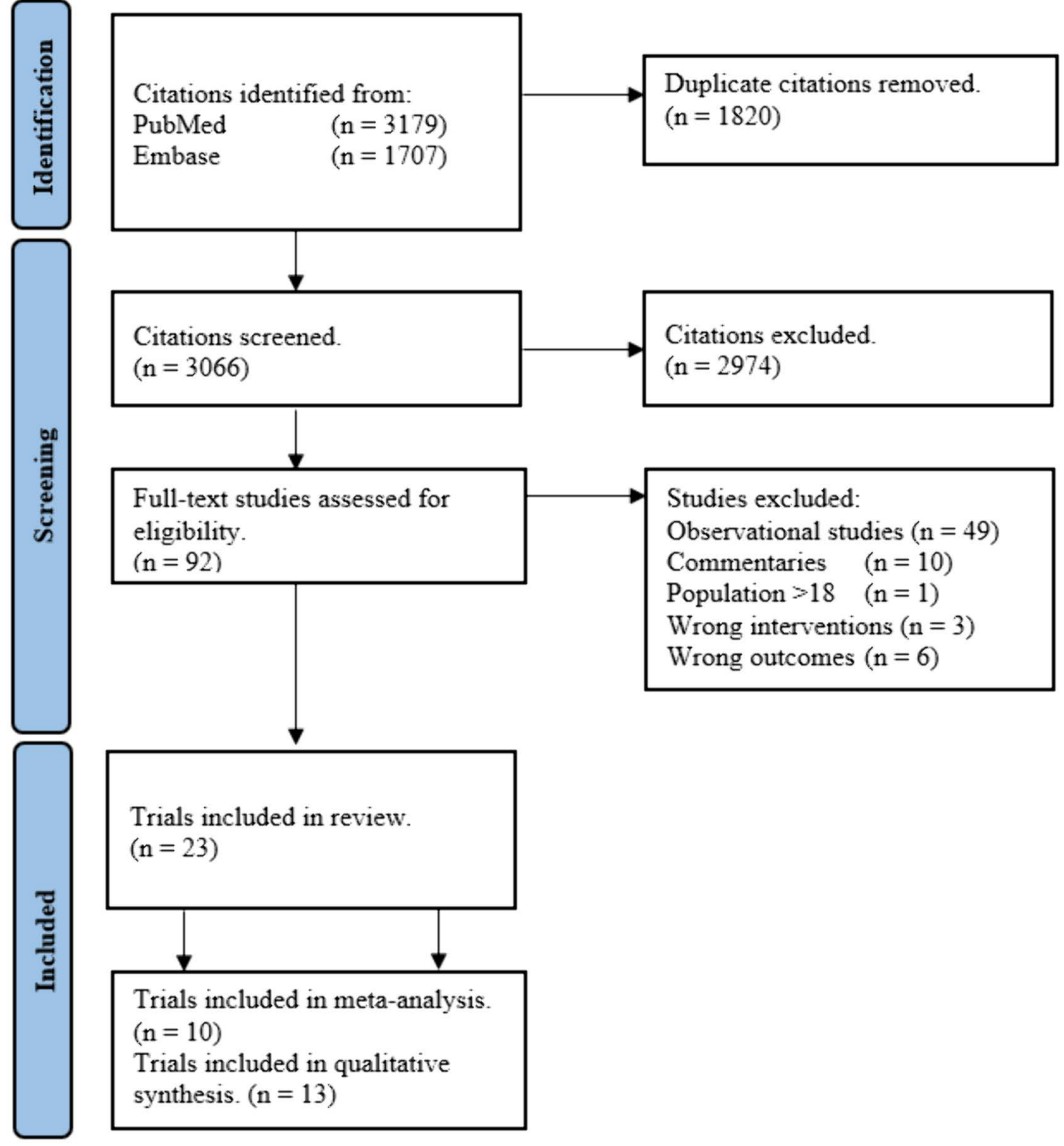

**Fig 1. PRISMA.** Flowchart of the selection process.

TSA included crude estimates from three original trials [11,12,16] comparing two-dose and one-dose measles vaccine schedules. Two trials were excluded due to non-comparable designs. One trial [23] used a 'MCV-for-all-policy' versus a 'restricted MCV. The other trial [22] used a composite outcome of both admissions and mortality. When comparing two vs. one MCV dose, the TSA showed that enough data was collected to conclusively reject a 25% relative reduction in the mortality risk, pooled effect RR = 1·05 (0·78 to 1·41), p = 0·76 (Fig 3). This is shown by the z-curve (Fig 3, green line). The z-curve falls below the 97·5 percentile of the standard normal distribution of 1·96 (Fig 3, blue dashed line), indicating no significant difference. The confidence interval and p-value lead to the same conclusion. Furthermore, the z-curve reached the area of futility. This is represented by the red triangle. This triangle is calculated by the TSA software and illustrates the point at which sufficient data has been included into the analysis to conclusively reject the hypothesis.

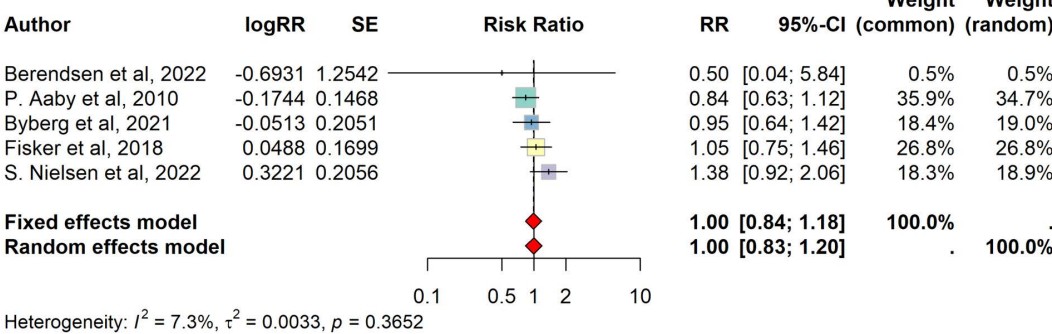

| Author | logRR | SE | Risk Ratio | RR | 95%-CI | Weight (common) | Weight (random) |
|---|---|---|---|---|---|---|---|
| Berendsen et al, 2022 | -0.6931 | 1.2542 | | 0.50 | [0.04; 5.84] | 0.5% | 0.5% |
| P. Aaby et al, 2010 | -0.1744 | 0.1468 | | 0.84 | [0.63; 1.12] | 35.9% | 34.7% |
| Byberg et al, 2021 | -0.0513 | 0.2051 | | 0.95 | [0.64; 1.42] | 18.4% | 19.0% |
| Fisker et al, 2018 | 0.0488 | 0.1699 | | 1.05 | [0.75; 1.46] | 26.8% | 26.8% |
| S. Nielsen et al, 2022 | 0.3221 | 0.2056 | | 1.38 | [0.92; 2.06] | 18.3% | 18.9% |
| **Fixed effects model** | | | | **1.00** | **[0.84; 1.18]** | **100.0%** | **.** |
| **Random effects model** | | | | **1.00** | **[0.83; 1.20]** | **.** | **100.0%** |

Heterogeneity: $I^2 = 7.3\%$, $\tau^2 = 0.0033$, $p = 0.3652$

**Fig 2. Forest plot: Mortality.** Forest plot of adjusted mortality estimates from original trials. Results shown for both fixed effect and random effects models.

### One vs zero doses of MCV

Three studies compared infants randomised to early MCV with infants randomised to no early dose as secondary analyses. All trials were original with non-overlapping populations. Mortality was compared after the early extra dose given at 4·5 months of age [11], 4 weeks after the third DTP vaccine [12], and at 4 months of age [16]), and at the routine dose at 9 months. The first trial [11] with follow-up of 4·5 months found mortality rate ratio (MRR) = 0·67 (0·38 to 1·19). The second trial [12] with follow-up from 4 weeks after the third DTP vaccine found MRR = 1·10 (0·66 to1·83). The third trial [16] with follow-up of 5 months found HR = 0·94 (0·45 to 1·96). No trial showed significant differences. Using crude estimates, a forest plot meta-analysis was performed. The fixed effect model resulted in a pooled RR = 0·90 (0·65 to 1·26). The random effects model yielded a pooled RR = 0·90 (0·65 to 1·26), I2 = 0% (Fig A in S2 Appendix). The TSA analysis found no significant difference for RRR = 25%, but more data were needed to reach a conclusive result (Fig B in S2 Appendix). For RRR = 33% (Fig C in S2 Appendix), the z-curve reached the area of futility. Thus, enough data was included to reach the conclusion that no 33% reduction in mortality risk was found when investigating one vs. zero doses of MCV.

### Potential sex-differential effects

Sex-specific subgroup analyses were performed. In males, a 25% mortality reduction was conclusively rejected, pooled effect RR = 1·09 (0·86 to 1·37) (Fig A in S3 Appendix). For females, the hypothesis of a 33% mortality reduction when comparing a two-dose programme with one dose was conclusively rejected, pooled effect RR = 1·0 (0·64 to 1·54) (Fig B in S3 Appendix).

### Morbidity

Three original trials (18,700 participants) investigated admissions or outpatient consultations as primary outcomes. One trial compared one MCV dose at 9 months with an extra booster dose at 18 months with follow-up until 48 months [22]. Another trial compared an early dose of MCV at 5–7 months with placebo and followed the children until 12 months [26]. The third original trial compared a dose of MCV vaccine with a health check-up and followed the children for 2 months [27]. Six other trials (26,519 participants) investigated these outcomes as secondary. Of these six trials, one was original. This trial compared a 'restrictive measles vaccine policy' with a 'measles vaccine for all policy' with follow-up until 60 months [23]. The remaining five trials were re-analyses or sub-studies of other trials included in this review [24,28–31].

The meta-analysis did not show any significant differences in admission rates between intervention and control groups across five trials with non-overlapping populations. Two forest plots were made using the crude and adjusted (for measles

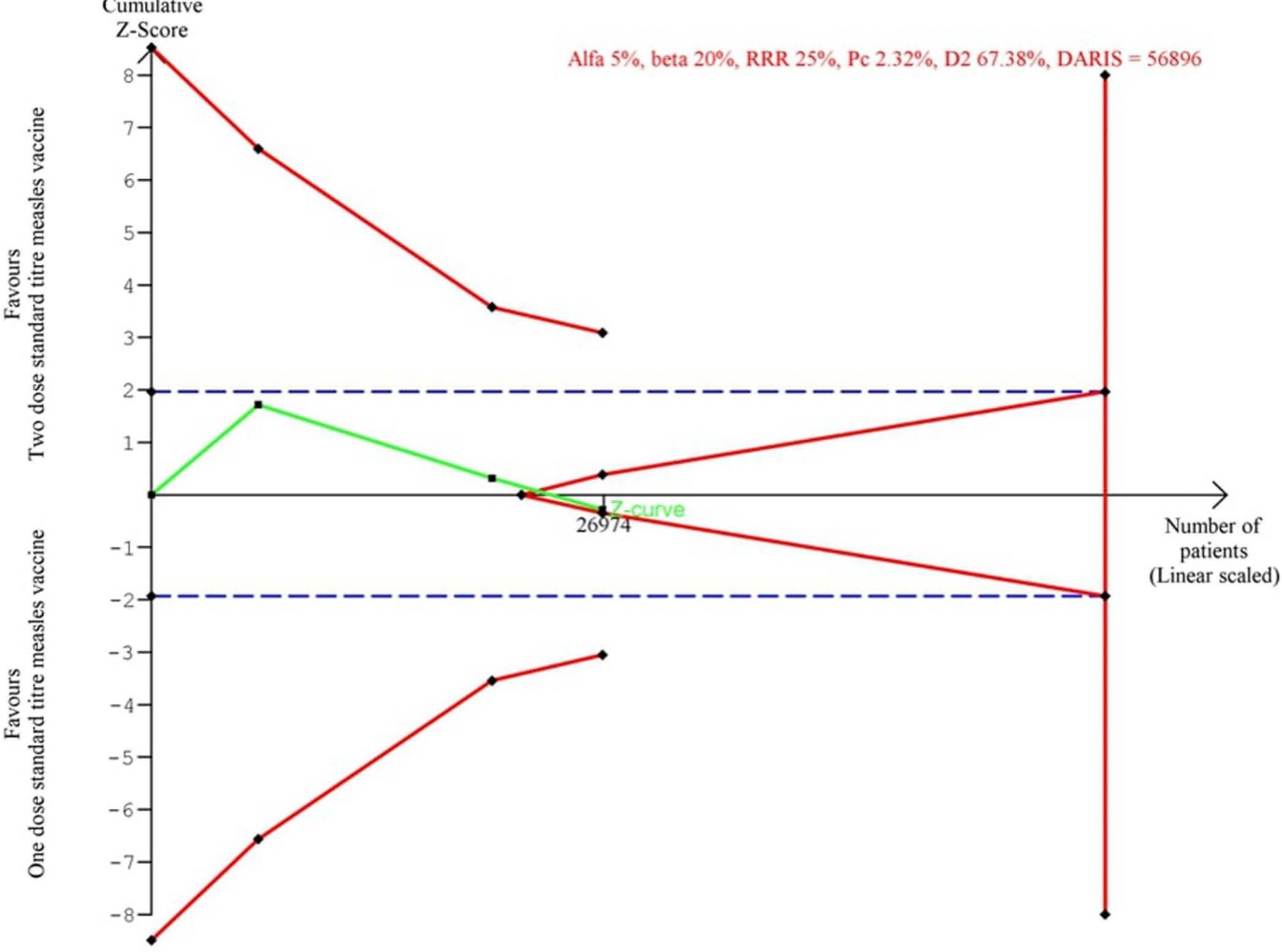

**Fig 3. TSA: Mortality.** Pooled Effect: RR = 1·05 (CI: 0·78 to 1·41, p = 0·76), Q = 5·71 (p = 0·06), I² = 0·65, D² = 0·67.

cases, accidental deaths, season, age, and sex) estimates of each trial. One trial was excluded since crude admission data were not published [22]. The pooled RR was 0·91 (0·82 to 1·00) (S3 Fig). Using adjusted estimates, the fixed effects model resulted in a pooled RR = 1·00 (0·92 to 1·08); the random effects model in RR = 1·00 (0·92 to 1·08), I2 = 1% (Fig 4).

Two trials with comparable interventions [24,31] were included in the TSA. The z-curve reached the area of futility, showing that enough data had been included to conclusively reject a 25% relative reduction in morbidity risk for two doses vs. one. Pooled effect RR = 0·94 (0·80 to 1·10), p = 0·43 (Fig 5).

### One vs zero doses of MCV

Three trials examined the effects of one versus zero doses of MCV. Two trials presented this as the primary analysis. The extra early dose was given at 4 weeks after the third DTP vaccine [24], at 5–7 months [26], and at 4.5 months [29]. Two trials were re-analyses of other RCTs [24, 29] and one was original [26]. No populations overlapped. The first trial reported

| Author | logRR | SE | Risk Ratio | RR | 95%-CI | Weight (common) | Weight (random) |
|--------|-------|-----|------------|-----|--------|-----------------|-----------------|
| Berendsen et al, 2022 | -0.3425 | 0.3428 | | 0.71 | [0.36; 1.39] | 1.5% | 1.5% |
| A. Varma et al, 2020 | -0.1744 | 0.1365 | | 0.84 | [0.64; 1.10] | 9.3% | 9.5% |
| M. Brønd et al, 2018 | -0.1508 | 0.1237 | | 0.86 | [0.67; 1.10] | 11.4% | 11.6% |
| Zimakoff et al 2023 | 0.0392 | 0.0663 | | 1.04 | [0.91; 1.18] | 39.6% | 39.2% |
| A. Schoeps et al, 2018 | 0.0488 | 0.0699 | | 1.05 | [0.92; 1.20] | 35.6% | 35.4% |
| Byberg et al, 2021 | 0.0862 | 0.2545 | | 1.09 | [0.66; 1.79] | 2.7% | 2.8% |
| **Fixed effect model** | | | | 1.00 | [0.92; 1.08] | 100.0% | . |
| **Random effects model** | | | | 1.00 | [0.92; 1.08] | . | 100.0% |

0.5　1　2

Heterogeneity: $I^2$ = 1.2%, $\tau^2$ = 0.0002, $p$ = 0.4084

**Fig 4. Forest plot: Morbidity.** Forest plot based on adjusted estimates from original trials of morbidity. Calculated using fixed and random effects models.

HR = 1·06 (0·77 to 1·45) [24], the second HR = 1·03 (0·91 to 1·18) [26], and the third HR = 0·78 (0·58 to 1·07) [29]. No significant differences were found. A forest plot using crude fixed effect estimates showed a pooled RR = 1·01 (0·93 to 1·09). The random effects model yielded a pooled RR = 0·93 (0·74 to 1·17), I2 = 68% (Fig A in S4 Appendix). The TSA analysis concluded that no statistical difference was found between the two groups based on a 25% RRR. (Fig B in S4 Appendix).

### Potential sex differential effects

A subgroup TSA by sex confirmed that a 25% relative reduction in mortality risk was conclusively rejected for both sexes when comparing two doses to one dose. In females, the pooled effect RR was 0·95 (0·77 to 1·18) (Fig A in S5 Appendix). In males, the pooled effect RR was 0·92 (0·83 to 1·03) (Fig B in S5 Appendix).

### The high titre measles vaccine

Eight trials or re-analyses on HTMV were identified [8,9,32–37]. One trial found an association between HTMV and a difference in mortality for both sexes. Three trials linked HTMV to increased mortality in female infants only. TSA from five trials [8,32,34,35,38] compared mortality effects of HTMV to STMV for both sexes. The z-curve crossed the boundary of harm (solid red line). Thus, a conclusive and significant negative effect of the HTMV was reproduced with a 25% risk reduction: RR = 1·22 (1·02 to 1·46, p = 0·03). (Fig A in S6 Appendix). The effect was significant in females only, but inconclusive at a 25% risk reduction as the z-curve did not reach the boundary of harm: RR = 1·45 (1·06 to 1·99, p = 0·02). (Fig B in S6 Appendix). TSA for females with a 33% RRR was also inconclusive, despite a significant negative effect (Fig C in S6 Appendix). More details on the HTMV can be found in S6 Appendix.

### Grading of recommendations, assessment, development, and evaluation & quality assessment

All articles were assessed for risk of bias using the ROB-2 tool (S2 Table). This evaluation was used in the completion of a GRADE assessment. GRADE assessments were made for each primary outcome in the protocol. Some trials investigated several outcomes and were included more than once in the GRADE table (S6-S7 Table).

The GRADE assessment for morbidity was low due to lack of blinding and the use of different interventions in the trials (S7 Table). For mortality, the lack of blinding of caregivers was not deemed a serious bias due to the indisputability of the outcome. Two GRADE assessments were made for mortality. One considered the overall body of evidence for mortality. Because this included both re-analyses, sub studies, and different intervention strategies, the overall quality rating was assessed to be low, reflecting the difficulty of drawing conclusions, despite the large amount of research in the field.

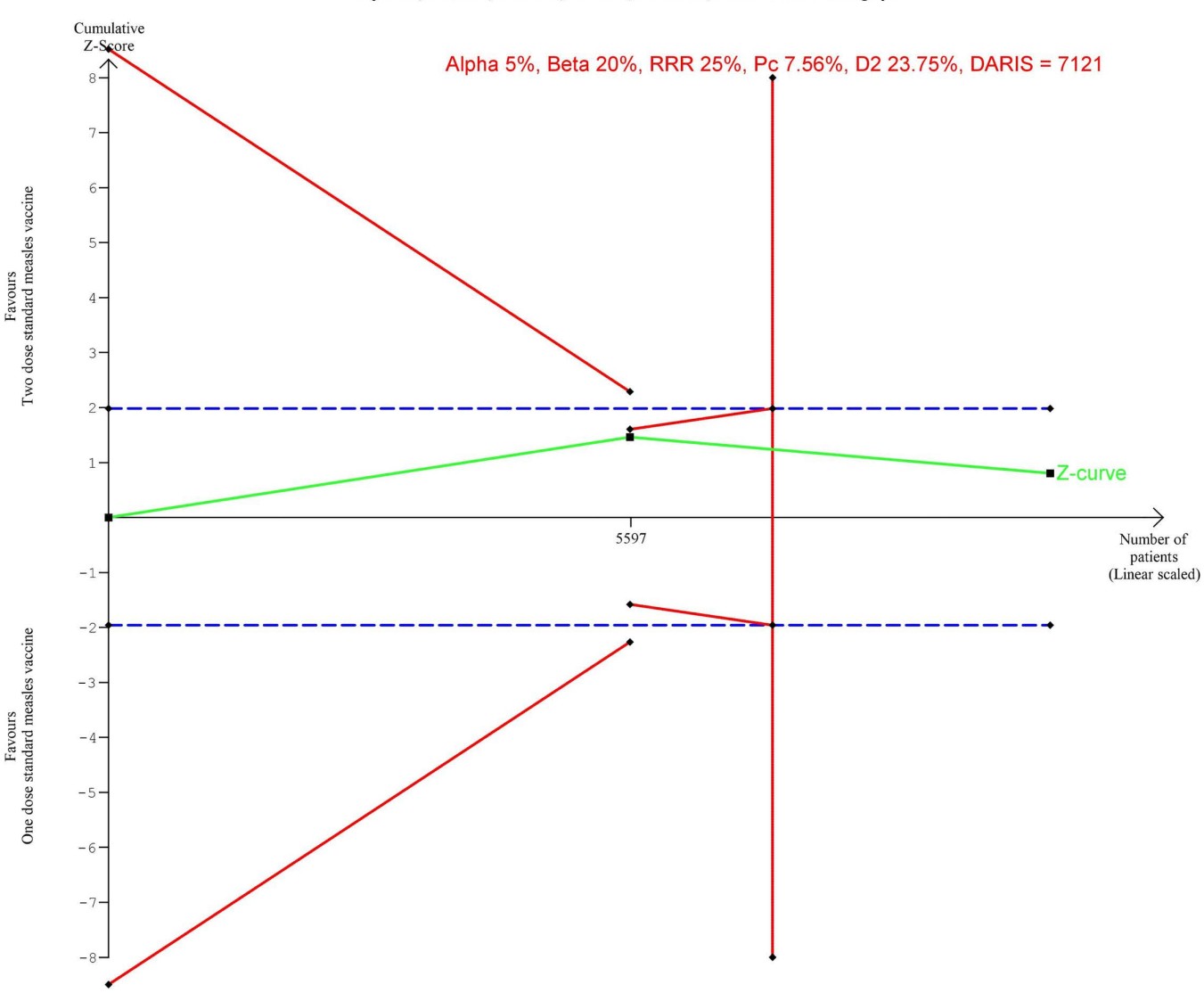

Alpha 5%, Beta 20%, RRR 25%, Pc 7.56%, D2 23.75%, DARIS is a Two-sided graph

Alpha 5%, Beta 20%, RRR 25%, Pc 7.56%, D2 23.75%, DARIS = 7121

**Fig 5. TSA: Morbidity.** Pooled Effect: RR = 0·94 (C.I: 0·80 to 1·10, p = 0·43), Q = 1·27 (p = 0·26), I² = 0·21, D² = 0·24.

However, during the review process, three large RCTs were identified and meta-analysed. These trials included a mortality outcome which reduced the impact of the lack of double blinding, had comparable interventions, and included 21,324 children in total. Thus, the GRADE assessment of these three RCTs alone was moderate to high (S7 Table). These trials were also applicable in a TSA. The z-curve for the TSA, based on these three trials, reached the area of futility, which leads to the conclusion of no non-specific effect on mortality after an extra early STMV dose.

## Discussion

In this systematic review and meta-analysis, we investigated whether measles-containing vaccines have beneficial non-specific effects on mortality and morbidity beyond protection against measles. In accordance with Cochrane standards, only randomised trials were included. Gold standard meta-analysis methods were applied. The analysis found no

support for beneficial non-specific effects of STMV. HTMV was linked to increased female mortality, but the result was not conclusive. Our findings align with a 2016 study by Higgins et al. [39] which found no such effects in randomised trials, but only in observational studies. This discrepancy was likely explained by healthy-and-wealthy-vaccinée bias [40,41] which is not present in randomised controlled trials [39].

## Standard-titre measles vaccines

Eight original trials showed no significant associations between MCV and morbidity or mortality. TSA based on three large, comparable trials found that enough data was included to draw this conclusion. The RRR level was 25% in the TSA to align with RRRs investigated in the included RCT's. Some upper confidence limits exceeded 1·25 in the forest plots, representing uncertainty in STMV effects. However, the TSA reached the area of futility, confirming that enough children were included to reject the hypothesised non-specific effects.

Three original RCT's [11,12,16] also presented secondary analyses with follow-up restricted to 9 months. This shorter follow-period represented the time between the first interventional early MCV and the second routine MCV and enabled comparison between zero versus one dose of MCV. The three RCT's were all large, with a combined population of 21,324 individuals, acceptable risks of bias, and highly comparable. None of the trials found a mortality difference between one and zero doses before nine months of age. The forest plot and TSA analysis showed no significant difference in mortality, and the area of futility was reached when RRR = 33% was investigated.

TSA showed that enough data was included to reject the hypothesis of non-specific effects in males (RRR = 25%) and in females (RRR = 33%).

For the outcome of morbidity, three RCTs [24,26,29], comprising one original and two re-analyses investigated a one vs zero dose intervention as a primary analysis. All RCTs were large with a combined population of 17,449 individuals. They had acceptable risks of bias and were highly comparable. None of these RCTs found a statistically significant difference in morbidity with a one vs zero dose intervention. The meta-analysis showed no significant difference in the forest plot and TSA, and this finding was conclusive in the TSA allowing us to conclude that there was no difference in morbidity between infants by sex, or by zero versus one dose of MCV.

## High titre measles vaccines

HTMVs were discontinued in 1992 by the World Health Organization due to reports on/concern regarding negative effects on long term non-measles related mortality [42]. The negative effect was corroborated by the meta-analysis strategy used throughout this review (S6 Appendix). This is likely a specific effect of the higher dose of attenuated measles virus [43]. Hence, this finding does not support the hypothesis of non-specific vaccine effects.

## Strengths and limitations

The strengths of the present systematic review were the gold standard meta-analytical approaches, including forest plots, and Trial Sequential Analysis. Only randomised controlled trials were included, and all analyses were pre-planned. Large populations were included in the meta-analyses, and an overview of all included trials and their re-analyses was created to ensure that meta-analyses did not include overlapping populations. The GRADE approach was used to assess the quality of evidence (S7 Table). Minor effects cannot be ruled out, but the RRR detection levels of the TSA were aligned with those of the included RCTs. Forest plots with both fixed and random effects were carried out for both crude and adjusted estimates. Only trials with a higher overall GRADE assessment were included in the TSA to minimise the heterogeneity and increase the quality of the results, which is a strength of this review. The statistical heterogenicity ($I^2$) was below 10% in all main meta-analyses.

Limitations included the fact that some trials lost many participants during follow-up, potentially leading to attrition bias or dependent censoring. No other missing data was found during the review process. Further, trials were included in the

overall meta-analysis irrespective of the assessed risk of bias. Many studies had a high risk of bias due to inadequate blinding necessitated by ethical considerations. Blinding was avoided to prevent caregivers from assuming their child had been vaccinated leading to missed vaccine doses. The limitation of lack of blinding was less serious for the indisputable outcome of mortality.

In some forest plots, wide confidence intervals were observed despite low heterogeneity estimates, potentially due to small sample sizes. Caution is warranted when interpreting p-values in analyses based on a small number of studies, as statistical power is reduced.

Another limitation was the utilisation of different interventions. As a trade-off between including all available RCT results and getting as close as possible to the true effect, different interventions were accepted in the forest plots. However, in the TSA, only fully comparable trials (with same intervention 'two dose vs. one dose' and 'one dose vs. zero doses regimes') were included. Neither the forest plots nor the TSAs revealed any sign of non-specific effects both if all available RCT's were included (as in the forest plots) and if only fully comparable RCT's were included (as in the TSAs).

Finally, due to the limited number of trials included, more formal assessments of publication bias (e.g., funnel plots and Egger's test) were not feasible. Thus, we did not conduct publication bias analyses. However, we note that if publication bias were present, – meaning studies finding no evidence of non-specific effects were less likely to be published – it would strengthen the conclusion that such an effect is absent, as the available evidence would be biased/skewed toward showing an effect.

## Conclusion

This systematic review and meta-analysis found no support for beneficial non-specific effects of STMV, but linked HTMV to increased female mortality.

## Supporting information

**S1 Appendix. Supplementary methods.**
(DOCX)

**S1 Fig. Overview of trial populations.** Shows how the included trials were categorised by intervention, outcome and study populations.
(DOCX)

**S1 File. Citations excluded with animal and language filters.**
(PDF)

**S1 Table. Systematic search string.**
(DOCX)

**S2 Table. Risk of bias assessment.**
(DOCX)

**S3 Table. Study characteristics and data included in main meta-analyses.**
(DOCX)

**S4 Table. Full data extraction including estimates and 95% confidence intervals.**
(DOCX)

**S5 Table. Exclusion reasons for all excluded studies after full text screening.**
(DOCX)

**S6 Table:  Decisions made in the TSA program in chronological order.**
(DOCX)

**S2 Fig.  Standard titre measles vaccine. Mortality. Two versus one doses. Crude data.**
(DOCX)

**S2 Appendix.  The standard titre measles vaccine. Mortality. One versus zero doses.**
(DOCX)

**S3 Appendix.  The standard titre measles vaccine. Mortality. Potential sex-differential effects.**
(DOCX)

**S3 Fig.  Morbidity. Standard titre measles vaccine. Two versus one doses. Crude data.**
(DOCX)

**S4 Appendix.  The standard titre measles vaccine. Morbidity. One versus zero doses.**
(DOCX)

**S5 Appendix.  The standard titre measles vaccine. Morbidity. Potential sex-differential effects.**
(DOCX)

**S6 appendix.  The high titre measles vaccine.**
(DOCX)

**S7 Table.  GRADE.**
(DOCX)

## Acknowledgments

The lead author (the manuscript's guarantor) affirms that the manuscript is an honest, accurate, and transparent account of the study being reported. No important aspects of the study have been omitted and any discrepancies from the study as planned have been explained. All authors attest they meet the ICMJE criteria for authorship.

## Author contributions

**Conceptualization:** Louise A Fournais, Anne C Zimakoff, Lone G Stensballe.

**Data curation:** Louise A Fournais, Anne C Zimakoff, Ingvild Fosse.

**Formal analysis:** Louise A Fournais, Andreas Jensen, Jeppe H Svanholm.

**Investigation:** Louise A Fournais, Anne C Zimakoff, Jeppe H Svanholm, Ingvild Fosse.

**Methodology:** Louise A Fournais, Anne C Zimakoff, Andreas Jensen, Jeppe H Svanholm, Ingvild Fosse, Lone G Stensballe.

**Project administration:** Louise A Fournais, Anne C Zimakoff, Jeppe H Svanholm, Lone G Stensballe.

**Resources:** Louise A Fournais.

**Software:** Louise A Fournais, Andreas Jensen, Jeppe H Svanholm.

**Supervision:** Louise A Fournais, Anne C Zimakoff.

**Validation:** Louise A Fournais, Anne C Zimakoff, Lone G Stensballe.

**Visualization:** Louise A Fournais, Lone G Stensballe.

**Writing – original draft:** Louise A Fournais.

**Writing – review & editing:** Louise A Fournais, Anne C Zimakoff, Andreas Jensen, Jeppe H Svanholm, Ingvild Fosse, Lone G Stensballe.

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
