## [Decision Letter · Decision Letter 0]

Dear Dr. Fournais,

Thank you for submitting your manuscript to PLOS ONE. After careful consideration, we feel that it has merit but does not fully meet PLOS ONE’s publication criteria as it currently stands. Therefore, we invite you to submit a revised version of the manuscript that addresses the points raised during the review process.

We look forward to receiving your revised manuscript.

Kind regards,

Ahmad Khalid Aalemi, M.D., M.Sc., Ph.D.

Academic Editor

PLOS ONE

Journal Requirements:

2. As required by our policy on Data Availability, please ensure your manuscript or supplementary information includes the following:

3. In the online submission form, you indicated that data sharing: Data and analyses will be available on request from the corresponding author (LAF).

5. Please remove all personal information, ensure that the data shared are in accordance with participant consent, and re-upload a fully anonymized data set

Additional Editor Comments :

Dear Louise,

Please revise your manuscript based on the reviewer comments.

Reviewers' comments:

Reviewer's Responses to Questions

**Comments to the Author**

1. Is the manuscript technically sound, and do the data support the conclusions?

Reviewer #1: Yes

Reviewer #2: Yes

Reviewer #3: Partly

Reviewer #4: Yes

2. Has the statistical analysis been performed appropriately and rigorously?

Reviewer #1: Yes

Reviewer #2: Yes

Reviewer #3: No

Reviewer #4: Yes

3. Have the authors made all data underlying the findings in their manuscript fully available?

Reviewer #1: Yes

Reviewer #2: Yes

Reviewer #3: Yes

Reviewer #4: Yes

4. Is the manuscript presented in an intelligible fashion and written in standard English?

Reviewer #1: Yes

Reviewer #2: Yes

Reviewer #3: Yes

Reviewer #4: Yes

Reviewer #1: As written, the introduction does not provide an adequate argument for the need for the study. The background should be very strong to persuade the reader. Need more references for citing in the discussion. The language of the article will become even better and more understandable with the revision.

Reviewer #2: Review the information between lines 311 to 315, there is some extra spaces and the information could be adjusted to stay in the same paragraph.

The manuscript is easy to read and the information is clear. I would suggest make the figures bigger, except the figure 1, and colorful to make the differences or not more visible.

Reviewer #3: Overall Assessment

This manuscript presents a systematic review and meta-analysis assessing the hypothesis that standard-titre measles vaccines (STMV) have beneficial non-specific effects on mortality and morbidity. The study follows PRISMA guidelines and includes Trial Sequential Analysis (TSA) to ensure robustness. The authors conclude that their meta-analysis of randomized controlled trials (RCTs) does not support the hypothesis of non-specific effects of STMV, while historical high-titre measles vaccines (HTMV) were associated with increased female mortality.

The manuscript is well-structured, follows rigorous methodological standards, and provides a valuable contribution to the ongoing discussion about the non-specific effects of vaccines. However, there are areas where improvements could be made, particularly in the depth of discussion, clarity in statistical reporting, and consideration of potential limitations.

⸻

Major Strengths

1. Rigorous Methodology and Meta-Analysis Approach

• The study adheres to PRISMA guidelines and follows Cochrane Handbook recommendations for systematic reviews.

• The use of TSA is a significant strength, ensuring that conclusions are drawn from an adequate amount of data.

• Inclusion criteria were stringent, only considering RCTs, which minimizes bias from observational studies.

2. Comprehensive Literature Review

• The authors conducted a systematic search in major databases (EMBASE and PubMed) and repeated it weekly for nearly a year.

• The screening process involved independent reviewers with a clear conflict-resolution strategy.

3. Robust Statistical Analysis

• Meta-analysis includes both fixed and random effects models, which appropriately account for heterogeneity.

• TSA provides a higher level of confidence in the conclusions by assessing whether sufficient data exist to confirm or reject the hypothesis.

• Sex-stratified analyses enhance the understanding of differential effects.

⸻

Major Weaknesses and Areas for Improvement

1. Interpretation of Findings

• The authors state that their meta-analysis “did not support the hypothesis of non-specific effects of STMV.” However, the confidence intervals in several analyses are wide, and the rejection of a 25% or 33% risk reduction does not rule out smaller potential benefits.

• The authors could discuss the implications of their findings in the context of potential smaller effects that might be clinically relevant but undetectable within the current dataset.

2. Statistical Considerations

• The TSA was performed using a predefined 25% RRR, but this threshold is somewhat arbitrary. The authors should justify why this specific reduction was chosen.

• While TSA helps assess whether sufficient data have been collected, some meta-analyses still showed confidence intervals that include potential small effects. This should be discussed more explicitly.

3. Discussion on Potential Biases

• While RCTs are considered the highest level of evidence, there are limitations:

• Many trials had substantial loss to follow-up, which can introduce attrition bias.

• The lack of blinding in many trials could have influenced results, particularly for morbidity outcomes where subjective reporting may be involved.

• Heterogeneity among studies, particularly regarding intervention timing and population characteristics, could have affected outcomes. This heterogeneity should be explored more thoroughly in the discussion.

4. Consideration of Alternative Explanations

• Some prior observational studies have reported non-specific effects of measles vaccines, whereas this meta-analysis does not find such effects in RCTs. The authors should explore why observational studies might have produced different results (e.g., confounding, healthy vaccinee bias).

• The review does not consider potential age-related immune system differences that might influence vaccine effects. Could earlier vaccination (before 9 months) have an impact that was not captured in the TSA?

⸻

Minor Issues and Suggestions

1. Clarity in Presentation

• Some of the statistical results in the text could be more clearly presented. For example, rather than just stating “TSA showed that enough data was collected to conclusively reject the hypothesis,” the authors should specify the exact thresholds and explain them in a more intuitive manner.

• Figures and tables should be better integrated into the discussion. The manuscript refers to supplementary figures often, but some key findings should be highlighted directly in the main text.

2. Ethical Considerations

• The manuscript states that ethical approval was not needed. However, it would be useful to briefly explain why, especially since some of the included studies involved human participants.

3. Formatting and Writing Style

• The manuscript is generally well-written, but some sentences are long and difficult to follow. Simplifying complex statements and breaking them into smaller sentences would improve readability.

⸻

Conclusion and Recommendation

This manuscript presents a well-executed systematic review and meta-analysis that rigorously evaluates the hypothesis of non-specific effects of measles vaccines. While the findings do not support a strong beneficial effect of STMV, the discussion could be improved by acknowledging the potential for smaller effects, addressing study limitations in more depth, and better justifying statistical choices.

Reviewer #4: This is a clearly written systematic review and meta-analysis of randomised controlled trials with both standard titre and high titre live attenuated measles containing vaccines as intervention and other vaccines or placebo as comparator. The primary outcomes were mortality and morbidity Other secondary outcomes were examined as well. The analysis was interesting in that the authors added the Trial Sequential Analysis (TSA) approach having some advantageous features which was explained in the supplement.

The systematic review was routine and well done in that the investigators used PRISMA and PROSPERO. The studies included were only RCT’s . The approach to the Systematic search and screening and Quality assessment, resolution of reviewer differences as well as data extraction were explained adequately. Risk of bias assessment and other such features were reasonably outlined in the supplemental material. Obviously, publication bias was not discussed as the number of articles was perhaps inadequate for such. This should be made known to the reader.

Analysis was presented with Forest plots and the usual efficacy and heterogeneity type statistics were included.

The analysis appeared to confirm that the present review and meta-analysis did not support the hypothesized beneficial non-specific effects of STMV and found the historical HTMV to be possibly associated with increased female mortality.

There are some minor statistical concerns. The limitations are adequately noted by the investigators . However, they should mention to the reader, as noted above, why the lack of publication bias discussion, unless this reviewer missed it. Also, they should note the small number of articles in Figures S9 and S11 prompting one to interpret any p-values with caution. Also, any discussion would be helpful of heterogeneity or other causes of unusually wide confidence bands on any of the Forest plots both in the manuscript and supplement.

**Do you want your identity to be public for this peer review?** For information about this choice, including consent withdrawal, please see our Privacy Policy

Reviewer #1: **Yes: ** DR. YAHYA H. Y. ALFARRA, BDS (Hons), MSc, PhD

Reviewer #2: No

Reviewer #3: **Yes: ** Mohamed Samy Abousenna

Reviewer #4: No

---

## [Author Response · Author response to Decision Letter 1]

7 May 2025

Author’s reply: Thank you. The manuscript has now been updated to fit the PLOS ONE style requirements and file naming.

2. As required by our policy on Data Availability, please ensure your manuscript or supplementary information includes the following:

Author’s reply: Thank you. A new table (S5 Table) has now been added to the supplementary materials showing all exclusion reasons for our 69 excluded articles after full text screening.

Author’s reply: No included studies are unpublished.

Author’s reply: Thank you. S4 Table has now been updated and includes name of data extractors and date.

Author’s reply: Thank you. S4 Table now includes a new column confirming whether each study was eligible for inclusion and whether each study was eligible for metaanalysis

Author’s reply: Thank you. All data needed to replicate our analyses can now be easily accessed in S4 Table.

Author’s reply: No data were from another source.

Author’s reply: A GRADE and a risk of bias assessment table has been included in supplementary materials. S2 Table and S7 Table.

Author’s reply: Thank you. Elaboration on missing data was included in the discussion l 321-323. PLEASE NOTE that all line numbers that are given in this document are based on looking at the manuscript with track changes. Track changes should be shown with ‘simple markup’ in word.

3. In the online submission form, you indicated that data sharing: Data and analyses will be available on request from the corresponding author (LAF).

Author’s reply: All data that were available on request will now be a part of supplementary material. See S1 file and S5 Table.

Author’s reply: Thank you. Affiliations have been updated to include both the affiliation where the work of this paper was primarily carried out, and the current affiliation.

5. Please remove all personal information, ensure that the data shared are in accordance with participant consent, and re-upload a fully anonymized data set

Author’s reply: Thank you. No personal information was included in the manuscript. No participant consent was obtained before the beginning of this paper since all data from this paper was already published in other papers that had the responsibility to obtain participant consent. This reasoning has now been added to the manuscript p. 15 l. 366-367

Author’s reply: Thank you. All supporting information files are now captioned at the end of the manuscript and in the correct chronological order. Furthermore in-text citations match. The supplementary files will be uploaded according to the provided guidelines.

Additional Editor Comments :

Dear Louise,

Please revise your manuscript based on the reviewer comments.

Reviewers' comments:

Reviewer's Responses to Questions

Comments to the Author

1. Is the manuscript technically sound, and do the data support the conclusions?

Reviewer #1: Yes

Reviewer #2: Yes

Reviewer #3: Partly

Reviewer #4: Yes

2. Has the statistical analysis been performed appropriately and rigorously?

Reviewer #1: Yes

Reviewer #2: Yes

Reviewer #3: No

Reviewer #4: Yes

3. Have the authors made all data underlying the findings in their manuscript fully available?

Reviewer #1: Yes

Reviewer #2: Yes

Reviewer #3: Yes

Reviewer #4: Yes

4. Is the manuscript presented in an intelligible fashion and written in standard English?

Reviewer #1: Yes

Reviewer #2: Yes

Reviewer #3: Yes

Reviewer #4: Yes

5. Review Comments to the Author

Reviewer #1: As written, the introduction does not provide an adequate argument for the need for the study. The background should be very strong to persuade the reader. Need more references for citing in the discussion. The language of the article will become even better and more understandable with the revision.

Author’s reply: Thank you for this valuable feedback. In the revised manuscript, the Background and Discussion sections have been elaborated upon accordingly.

Reviewer #2: Review the information between lines 311 to 315, there is some extra spaces and the information could be adjusted to stay in the same paragraph.

Author’s reply: Thank you. The text was revised accordingly.

The manuscript is easy to read and the information is clear. I would suggest make the figures bigger, except the figure 1, and colorful to make the differences or not more visible.

Author’s reply: Thank you. All supplementary figures have been made bigger and/or more colourful. Colors have been added to all figures in the manuscript. But the figures in the main text must comply with the requirements from PLOS ONE and therefore this will dictate the size of these figures.

Reviewer #3: Overall Assessment

This manuscript presents a systematic review and meta-analysis assessing the hypothesis that standard-titre measles vaccines (STMV) have beneficial non-specific effects on mortality and morbidity. The study follows PRISMA guidelines and includes Trial Sequential Analysis (TSA) to ensure robustness. The authors conclude that their meta-analysis of randomized controlled trials (RCTs) does not support the hypothesis of non-specific effects of STMV, while historical high-titre measles vaccines (HTMV) were associated with increased female mortality.

The manuscript is well-structured, follows rigorous methodological standards, and provides a valuable contribution to the ongoing discussion about the non-specific effects of vaccines. However, there are areas where improvements could be made, particularly in the depth of discussion, clarity in statistical reporting, and consideration of potential limitations.

⸻

Major Strengths

1. Rigorous Methodology and Meta-Analysis Approach

• The study adheres to PRISMA guidelines and follows Cochrane Handbook recommendations for systematic reviews.

• The use of TSA is a significant strength, ensuring that conclusions are drawn from an adequate amount of data.

• Inclusion criteria were stringent, only considering RCTs, which minimizes bias from observational studies.

2. Comprehensive Literature Review

• The authors conducted a systematic search in major databases (EMBASE and PubMed) and repeated it weekly for nearly a year.

• The screening process involved independent reviewers with a clear conflict-resolution strategy.

3. Robust Statistical Analysis

• Meta-analysis includes both fixed and random effects models, which appropriately account for heterogeneity.

• TSA provides a higher level of confidence in the conclusions by assessing whether sufficient data exist to confirm or reject the hypothesis.

• Sex-stratified analyses enhance the understanding of differential effects.

⸻

Major Weaknesses and Areas for Improvement

1. Interpretation of Findings

• The authors state that their meta-analysis “did not support the hypothesis of non-specific effects of STMV.” However, the confidence intervals in several analyses are wide, and the rejection of a 25% or 33% risk reduction does not rule out smaller potential benefits.

• The authors could discuss the implications of their findings in the context of potential smaller effects that might be clinically relevant but undetectable within the current dataset.

Author’s reply: Thank you. In the Discussion of the revised manuscript, this aspect was clarified. l. 314-317.

2. Statistical Considerations

• The TSA was performed using a predefined 25% RRR, but this threshold is somewhat arbitrary. The authors should justify why this specific reduction was chosen.

Author’s reply: Thank you. As also mentioned in the manuscript, the RRRs of the TSAs were consequently aligned with those of the included RCTs. In the Discussion of the revised manuscript, this aspect was elaborated upon. l. 278

• While TSA helps assess whether sufficient data have been collected, some meta-analyses still showed confidence intervals that include potential small effects. This should be discussed more explicitly.

Author’s reply: Thank you. In the Discussion of the revised manuscript, this aspect was clarified. l. 329-331

3. Discussion on Potential Biases

• While RCTs are considered the highest level of evidence, there are limitations:

• Many trials had substantial loss to follow-up, which can introduce attrition bias.

Author’s reply: Thank you. In the Limitations section of the Discussion of the revised manuscript, attrition bias was elaborated upon lines 321-323.

• The lack of blinding in many trials could have influenced results, particularly for morbidity outcomes where subjective reporting may be involved.

Author’s reply: Thank you. In the Limitations section of the Discussion of the revised manuscript, lack of blinding was elaborated upon lines 323-327.

• Heterogeneity among studies, particularly regarding intervention timing and population characteristics, could have affected outcomes. This heterogeneity should be explored more thoroughly in the discussion.

Author’s reply: Thank you. In the Limitations section of the Discussion of the revised manuscript, the important aspect of heterogeneity, which was measured to be below 10% throughout the present systematic review, was elaborated upon lines 319-320.

4. Consideration of Alternative Explanations

• Some prior observational studies have reported non-specific effects of measles vaccines, whereas this meta-analysis does not find such effects in RCTs. The authors should explore why observational studies might have produced different results (e.g., confounding, healthy vaccinee bias).

Author’s reply: Thank you. The importance of confounding and healthy-and-wealthy-vaccinée bias (inclusive relevant references) have been added to the Discussion of the revised manuscript. Lines 270-273

• The review does not consider potential age-related immune system differences that might influence vaccine effects. Could earlier vaccination (before 9 months) have an impact that was not captured in the TSA?

Author’s reply: Thank you for this relevant question, which was addressed in the 0 vs. 1 dose of MCV analysis described separately in the results section. Also for early MCV administration, no non-specific vaccine effects were observed.

Minor Issues and Suggestions

1. Clarity in Presentation

• Some of the statistical results in the text could be more clearly presented. For example, rather than just stating “TSA showed that enough data was collected to conclusively reject the hypothesis,” the authors should specify the exact thresholds and explain them in a more intuitive manner.

Author’s reply: Thank you. A more intuitive explanation of thresholds and impact of the figures has been added to the manuscript when the first TSA fig 2 is presented. Lines 153-159.

• Figures and tables should be better integrated into the discussion. The manuscript refers to

---

## [Decision Letter · Decision Letter 1]

Measles vaccines and non-specific effects on mortality or morbidity. A systematic review and meta-analysis.

PONE-D-24-29586R1

Dear Dr. Louis Amstrup Fournis,

We’re pleased to inform you that your manuscript has been judged scientifically suitable for publication and will be formally accepted for publication once it meets all outstanding technical requirements.

Kind regards,

Ahmad Khalid Aalemi, M.D., M.Sc., Ph.D.

Academic Editor

PLOS ONE

Additional Editor Comments (optional):

Reviewers' comments:

Reviewer's Responses to Questions

**Comments to the Author**

Reviewer #2: All comments have been addressed

Reviewer #3: All comments have been addressed

Reviewer #4: All comments have been addressed

2. Is the manuscript technically sound, and do the data support the conclusions?

Reviewer #2: Yes

Reviewer #3: Yes

Reviewer #4: (No Response)

3. Has the statistical analysis been performed appropriately and rigorously?

Reviewer #2: Yes

Reviewer #3: Yes

Reviewer #4: (No Response)

4. Have the authors made all data underlying the findings in their manuscript fully available?

Reviewer #2: Yes

Reviewer #3: Yes

Reviewer #4: (No Response)

5. Is the manuscript presented in an intelligible fashion and written in standard English?

Reviewer #2: Yes

Reviewer #3: Yes

Reviewer #4: (No Response)

Reviewer #2: (No Response)

Reviewer #3: Thank you for your thoughtful and thorough revision of the manuscript titled "Measles vaccines and nonspecific effects on mortality or morbidity. A systematic review and meta-analysis." I appreciate the effort you made to address each of the points raised during the initial review.

Reviewer #4: (No Response)

**Do you want your identity to be public for this peer review?** For information about this choice, including consent withdrawal, please see our Privacy Policy

Reviewer #2: No

Reviewer #3: **Yes: ** Mohamed Samy Abousenna

Reviewer #4: No

---

## [Editor Report · Acceptance letter]

PONE-D-24-29586R1

PLOS ONE

Dear Dr. Fournais,

I'm pleased to inform you that your manuscript has been deemed suitable for publication in PLOS ONE. Congratulations! Your manuscript is now being handed over to our production team.

Kind regards,

on behalf of

Dr. Ahmad Khalid Aalemi

Academic Editor

PLOS ONE